

# Percentile curves for skinfold thickness for Canadian children and youth

Stefan Kuhle[1], Jillian Ashley-Martin[1], Bryan Maguire[2] and David C. Hamilton[2]

[1] Departments of Pediatrics and Obstetrics & Gynaecology, Dalhousie University, Halifax, NS, Canada
[2] Department of Mathematics & Statistics, Dalhousie University, Halifax, NS, Canada

## ABSTRACT

**Background.** Skinfold thickness (SFT) measurements are a reliable and feasible method for assessing body fat in children but their use and interpretation is hindered by the scarcity of reference values in representative populations of children. The objective of the present study was to develop age- and sex-specific percentile curves for five SFT measures (biceps, triceps, subscapular, suprailiac, medial calf) in a representative population of Canadian children and youth.

**Methods.** We analyzed data from 3,938 children and adolescents between 6 and 19 years of age who participated in the Canadian Health Measures Survey cycles 1 (2007/2009) and 2 (2009/2011). Standardized procedures were used to measure SFT. Age- and sex-specific centiles for SFT were calculated using the GAMLSS method.

**Results.** Percentile curves were materially different in absolute value and shape for boys and girls. Percentile girls in girls steadily increased with age whereas percentile curves in boys were characterized by a pubertal centered peak.

**Conclusions.** The current study has presented for the first time percentile curves for five SFT measures in a representative sample of Canadian children and youth.

## INTRODUCTION

The rising prevalence of overweight and obese children and associated public health toll in Canada and other developed countries is well established (*Shields, 2006*; *Tran et al., 2013*; *Ng et al., 2014*). Effective obesity prevention and treatment efforts require reliable identification of the at risk population. Specifically, accurate characterization of childhood body composition is essential for identifying children who exceed recommended weight norms or may be at risk of future excess weight and related cardiovascular and metabolic health conditions. Though body mass index is the most commonly used method for assessing childhood body composition, it does not provide an accurate estimate of adiposity (*Frankenfield et al., 2001*; *Brambilla et al., 2006*). Childhood adiposity is potentially more strongly associated with future body composition and metabolic status than childhood BMI (*Freedman et al., 1999*; *Nooyens et al., 2007*). Childhood adiposity is also positively associated with certain cardiovascular and metabolic disease risk factors (*Going et al., 2011*; *Dai et al., 2009*). Skinfold thickness (SFT) measures are a feasible and reliable estimate of body fat (*Boeke et al., 2013*; *Sardinha et al., 1999*; *Bedogni et al., 2003*), and have been shown to be predictive of elevated levels of cardiovascular disease risk factors (*Steinberger et*

Corresponding author
Stefan Kuhle, stefan.kuhle@dal.ca

*al., 2005; Petkeviciene et al., 2015*) and metabolic syndrome (*Laurson, Eisenmann & Welk, 2011*). Interpretation and uptake of SFT measurement as a method for the assessment of body fat is hindered by the lack of reference data. While there are health-related cutoffs for BMI (*Cole et al., 2000; De Onis et al., 2007*), waist circumference (*World Health Organization, 2008*), and waist-to-height ratio (*McCarthy & Ashwell, 2006*), there is no comparable definition based on SFT for either children or adults. Percentile curves have been developed for US (*Laurson, Eisenmann & Welk, 2011; Addo & Himes, 2010*) and European (*Moreno et al., 2007; Heude et al., 2006; Haas, Liepold & Schwandt, 2011; Brannsether et al., 2013; Kromeyer-Hauschild, Glässer & Zellner, 2012; Jaworski et al., 2012; Wohlfahrt-Veje et al., 2014; Nagy et al., 2014*) children. The applicability of these values to the Canadian population is limited due to differences in childhood overweight and obesity prevalence. Moreover, previous SFT references have either been developed with a limited number of skinfolds (*Laurson, Eisenmann & Welk, 2011; Addo & Himes, 2010; Brannsether et al., 2013; Kromeyer-Hauschild, Glässer & Zellner, 2012*) or were based on a narrower age range (*Moreno et al., 2007; Haas, Liepold & Schwandt, 2011; Klimek-Piotrowska et al., 2015*). Therefore, the objective of the present study was to develop age- and sex-specific percentile curves for five SFT measures (biceps, triceps, subscapular, suprailiac, medial calf) in a representative population of Canadian children and youth.

## METHODS

The present study used data from the Canadian Health Measures Survey (CHMS) cycles 1 and 2. The CMHS is a representative, cross-sectional survey that assesses indicators of health and wellness in Canadians between 3 and 79 years (*Statistics Canada, 2011; Statistics Canada, 2012*). The survey consists of a household interview to obtain sociodemographic and health information, and a visit to a mobile examination centre to perform a number of physical measurements and tests. The sampling frame of the Canadian Labour Force Survey was used to identify the collection sites for the mobile examination centres. Within each collection site, households were selected using the 2006 Census as the sampling frame. Interviews and examinations for the CHMS Cycle 1 and 2 were performed between 2007 and 2009, and 2009 and 2011, respectively. The overall response rate in the two cycles was 51.7% and 55.7%, respectively. Data from the two cycles was combined as per Statistics Canada guidelines (*Statistics Canada, 2013*) and weighted to account for the design effect and non-response bias (*Statistics Canada, 2013*). A total of 11,999 persons participated in the physical examination part of the survey. The present analysis uses data from 3,938 children and adolescents (1,996 males and 1,942 females) between the ages of 6 and 19 years.

### Anthropometric measures

Body mass index was calculated from measured weight and height using the formula weight/height$^2$ (kg/m$^2$). Weight was measured using a calibrated digital scale (Mettler Toledo, Mississauga, ON, Canada) to the nearest 0.1 kg. Standing height was measured using a fixed stadiometer with a vertical backboard and a moveable headboard to the nearest 0.01 cm. Weight status (underweight, normal weight, overweight, obese) was determined based on the IOTF (International Obesity Task Force) growth reference (*Cole & Lobstein, 2012*).

PeerJ ——————————————————————————————————

All SFT measurements were performed by trained health professionals at the mobile examination centres using a Harpenden skinfold caliper to the nearest 0.2 mm. Each SFT was measured three times and the average of the three measurements was used. Triceps SFT was measured on the midline of the back of the arm at the mid-point level between the acromium process and the tip of the olecranon process. Biceps SFT was measured over the biceps at the same level as the midpoint for the triceps. Subscapular SFT was measured below the inferior angle of the scapula at an angle of 45 degrees to the spine. Suprailiac SFT was measured in the mid-axillary line above the crest of the ilium. Medial calf SFT was measured at the medial side of the calf at the point of the largest circumference. SFT measurements were not done on individuals with a BMI $\geq$ 30 kg/m$^2$.

## Statistical analysis

The data were split by sex and modeled using a four parameter $(\mu, \sigma, \nu, \tau)$ Box–Cox power exponential distribution (*Rigby & Stasinopoulos, 2004*). The GAMLSS method is an extension of the LMS method (*Cole & Green, 1992*) and assumes that when the data ($Y$) is transformed using the transformation:

$$z = \frac{(y/\mu)^\nu - 1}{\nu\sigma} \quad \nu \neq 0$$

$$z = \frac{\log_e(y/\mu)}{\sigma} \quad \nu = 0.$$

$Z$ follows a standard power exponential distribution with power parameter $\tau$.

The age-specific distribution expresses the mean, coefficient of variation, skewness, and kurtosis as parameters that change smoothly as a function of age by modeling them as cubic splines. These functions can be plotted as smooth curves in terms of age and are referred to as the $\mu$ (mean), $\sigma$ (variance), $\nu$ (skewness), and $\tau$ (kurtosis) curves. Centiles for a particular age are computed by using the values of the four parameters for the corresponding age. The 3rd, 10th, 25th, 50th, 75th, 90th, and 97th centile curves were computed for biceps, triceps, subscapular, suprailiac, and medial calf SFT.

To avoid unusual behaviours of the spline functions near the end of the age range, data from respondents up to age 30 years were used to fit the models. This modification produced smoother curves that more accurately reflect the population characteristics. Residual quantile plots ("worm plots") (*Van Buuren & Fredriks, 2001*) were used to assess the goodness of fit of each component of the models.

All calculations were performed using the sampling weights provided by *Statistics Canada (2013)* to account for design effect and non-response bias. The CHMS uses a multistage sampling design with two sampling frames to select its sample. The probability of an individual to be selected for the survey is determined as the product of the probability of selection at each stage. To correct for non-response, the weight of non-respondent households and individuals is redistributed to respondents within homogeneous response groups based on characteristics that are available for both respondents and non-respondents as determined from the Census of Canada (such as dwelling type or household income). A detailed description of the weighting procedure can be found elsewhere (*Statistics Canada, 2012*).

The statistical software package R (*R Core Team, 2016*) with the *gamlss* package (*Rigby & Stasinopoulos, 2006*) was used to perform the statistical analyses.

### Ethics

All processes used for cycles 1 and 2 of the CHMS were reviewed and approved by the Health Canada Research Ethics Board to ensure that internationally recognized ethical standards for human research were met and maintained. Written informed consent was obtained from all participants aged 14 years and older; parents or guardians gave consent on behalf of children aged 6–13 years, while the child provided his or her assent to participate (*Statistics Canada, 2011*; *Statistics Canada, 2012*). The current project was approved by the IWK Health Centre Research Ethics Board, Halifax, NS, Canada (File #1014413).

## RESULTS

Characteristics of the sample are shown in Table 1. The median and interquartile range for the five SFT measurements by age and sex are shown in Table 2. The parameter values $(\mu, \sigma, \nu, \tau)$ as well as the 3rd, 10th, 25th, 50th, 75th, 90th, and 97th percentiles for the SFT curves are presented by age and sex (Tables 3–7). Model diagnostics showed an adaequate fit for all models.

Percentile curves are materially different in both absolute values and shape for boys and girls (Figs. 1–5). Girls have higher median skinfold thickness than boys at all measurement sites (Table 2). All skinfold thickness measurements among girls are characterized by a relatively steady increase from childhood through adolescence despite differing absolute percentile values and rates of yearly change. Lower body (medial calf, suprailiac) skinfold thickness measurements steadily rise until adolescence at which point the rate of yearly increase diminishes. Among upper body measurements, the biceps percentile curve plateaus in early adolescence, whereas the triceps and subscapular curves steadily increase from age 6 to 19. No substantial differences in truncal (subscapular, suprailiac) and peripheral (triceps, biceps, calf) percentile curves among girls were observed.

Skinfold thickness curves in boys are characterized by a peak around age 12 years. The magnitude of this pubertal centered peak was most notable in the percentiles exceeding the median. Subsequent to the post-pubertal peak, skinfold thickness decreased in the peripheral measures (biceps, calf, triceps) and moderately increased in the truncal measures. There were no apparent distinguishing characteristics between the upper and lower body percentile curves in boys.

## DISCUSSION

The current study has presented for the first time percentile curves for five SFT measures based on a representative sample of Canadian children and youth aged 6–19 years. The percentile curves presented are meant to be descriptive rather than prescriptive as associations with cardiovascular disease markers or outcomes were not assessed. The data may be used by researchers as reference data for future studies.

Our findings are comparable with other studies that have examined the development of SFT in childhood and adolescence. Both the steady upward trend in girls and the pubertal

**Table 1** Characteristics of 4,115 Canadian children and youth aged 6–19 years in the Canadian Health Measures Survey cycles 1 and 2.

|  | Prevalence (%) |
|---|---|
| **Sex** | |
| Male | 51.5 |
| Female | 48.5 |
| **Region of Canada** | |
| Atlantic Canada | 6.7[a] |
| Québec | 22.5 |
| Ontario | 40.9 |
| Prairies | 17.8 |
| British Columbia | 12.1 |
| **Racial origin** | |
| White | 83.3 |
| Black | 6.3[a] |
| Asian | 8.1 |
| Other | 2.3[a] |
| **Weight status** | |
| Underweight | 7.2 |
| Normal weight | 66.2 |
| Overweight | 17.0 |
| Obese | 9.6 |
| **Household education** | |
| Secondary school or less | 14.1 |
| College | 50.2 |
| University | 35.7 |
| **Household income** | |
| $30,000 or less | 13.6 |
| $30,001–$60,000 | 23.3 |
| $60,001–$80,000 | 19.4 |
| $80,001–$100,000 | 16.6 |
| >$100,000 | 27.1 |

Notes.
[a]Coefficient of variation between 16.6% and 33.3%; interpret with caution as per Statistics Canada sampling variability reporting guidelines.

peak in boys were also observed in US (*Addo & Himes, 2010*), German (*Haas, Liepold & Schwandt, 2011*; *Kromeyer-Hauschild, Glässer & Zellner, 2012*; *Neuhauser et al., 2011*), Polish (*Jaworski et al., 2012*), and Norwegian children (*Brannsether et al., 2013*). Of note, the pubertal peak was less pronounced in samples with a narrower age ranges (*Moreno et al., 2007*; *Haas, Liepold & Schwandt, 2011*; *Brannsether et al., 2013*; *Kromeyer-Hauschild, Glässer & Zellner, 2012*). The absolute SFT values in our study were largely comparable to US data of 32,783 children ages 1–19 years collected between 1963–1994 (*Addo & Himes, 2010*): median triceps and subscapular SFT at age 12 years were comparable between girls in the CHMS (triceps: 13.5 mm; subscapular: 8.8 mm) and the US study (triceps: 13.1
**Table 2 Sample size, median, and interquartile range for biceps, triceps, subscapular, suprailiac, and medial calf skinfold thickness (mm) for Canadian children and youth aged 6–19 years.**

| Sex | Age (years) | n | Biceps | | Triceps | | Subscapular | | Suprailiac | | Medial calf | |
|---|---|---|---|---|---|---|---|---|---|---|---|---|
| | | | Median | IQR | Median | IQR | Median | IQR | Median | IQR | Median | IQR |
| Female | 6 | 154 | 5.0 | 1.6 | 10.5 | 3.1 | 5.4 | 2.3 | 5.9 | 2.7 | 8.7 | 3.3 |
| | 7 | 140 | 5.1 | 2.9 | 11.0 | 5.0 | 5.6 | 2.7 | 6.8 | 5.1 | 10.0 | 4.3 |
| | 8 | 164 | 6.4 | 4.3 | 11.9 | 6.8 | 6.6 | 6.0 | 8.1 | 7.7 | 10.5 | 7.0 |
| | 9 | 174 | 7.3 | 4.4 | 13.1 | 7.7 | 8.1 | 8.4 | 11.2 | 11.0 | 12.1 | 7.3 |
| | 10 | 193 | 6.9 | 3.4 | 13.1 | 6.4 | 8.3 | 6.4 | 10.6 | 10.0 | 12.7 | 7.3 |
| | 11 | 209 | 7.2 | 4.1 | 12.4 | 6.3 | 8.5 | 4.9 | 11.9 | 10.3 | 13.2 | 7.0 |
| | 12 | 127 | 7.5 | 4.1 | 13.9 | 6.4 | 8.9 | 7.7 | 13.6 | 11.9 | 12.7 | 9.4 |
| | 13 | 131 | 7.3 | 3.1 | 14.0 | 7.9 | 9.1 | 8.3 | 16.1 | 14.7 | 13.6 | 10.0 |
| | 14 | 116 | 7.8 | 4.1 | 16.1 | 8.8 | 12.0 | 7.4 | 17.3 | 12.3 | 15.9 | 9.4 |
| | 15 | 118 | 7.4 | 4.3 | 16.3 | 8.0 | 11.1 | 8.5 | 17.7 | 14.4 | 15.0 | 9.5 |
| | 16 | 109 | 7.0 | 1.8 | 17.0 | 4.5 | 10.6 | 5.4 | 18.9 | 8.9 | 15.9 | 7.1 |
| | 17 | 111 | 7.5 | 3.5 | 16.8 | 6.1 | 12.9 | 7.7 | 21.1 | 12.3 | 17.0 | 8.7 |
| | 18 | 104 | 7.4 | 2.7 | 17.3 | 5.1 | 11.8 | 8.3 | 19.4 | 10.4 | 14.8 | 7.0 |
| | 19 | 92 | 7.2 | 4.2 | 17.5 | 7.9 | 11.8 | 6.0 | 19.8 | 11.2 | 16.9 | 8.1 |
| Male | 6 | 152 | 4.3 | 2.6 | 9.0 | 3.6 | 5.0 | 2.4 | 5.3 | 3.4 | 7.7 | 4.2 |
| | 7 | 163 | 5.0 | 3.6 | 10.2 | 7.0 | 5.3 | 5.3 | 5.6 | 8.1 | 9.1 | 7.0 |
| | 8 | 167 | 5.2 | 3.6 | 10.4 | 5.9 | 6.1 | 3.8 | 7.2 | 8.5 | 8.2 | 5.6 |
| | 9 | 164 | 6.0 | 4.8 | 11.1 | 7.5 | 6.2 | 5.3 | 7.0 | 10.3 | 9.8 | 9.1 |
| | 10 | 204 | 6.8 | 5.8 | 12.8 | 8.8 | 7.5 | 9.4 | 9.6 | 15.2 | 12.4 | 11.5 |
| | 11 | 185 | 5.5 | 4.5 | 11.1 | 8.8 | 6.8 | 4.8 | 9.9 | 9.6 | 10.3 | 9.1 |
| | 12 | 148 | 5.7 | 4.4 | 12.1 | 8.4 | 6.8 | 5.0 | 7.9 | 13.4 | 10.6 | 9.3 |
| | 13 | 141 | 5.3 | 4.6 | 10.8 | 8.3 | 7.0 | 5.7 | 9.7 | 10.4 | 10.5 | 9.8 |
| | 14 | 136 | 4.4 | 2.3 | 9.0 | 3.1 | 7.3 | 2.4 | 9.8 | 5.5 | 9.5 | 5.7 |
| | 15 | 119 | 4.4 | 2.2 | 8.1 | 5.0 | 7.2 | 2.3 | 9.2 | 5.2 | 7.9 | 5.7 |
| | 16 | 130 | 4.0 | 1.8 | 8.2 | 4.9 | 7.8 | 2.9 | 10.2 | 6.7 | 7.9 | 5.5 |
| | 17 | 114 | 3.9 | 1.9 | 8.4 | 4.6 | 8.5 | 3.8 | 10.2 | 11.5 | 8.4 | 5.3 |
| | 18 | 91 | 4.2 | 3.3 | 8.8 | 5.7 | 9.4 | 4.6 | 13.2 | 12.9 | 7.4 | 8.5 |

**Notes.**

IQR, Interquartile range.

mm; subscapular: 8.2 mm). Median triceps SFT at age 12 in CHMS boys was slightly lower than reported in US boys (11.3 mm vs. 13.1 mm) whereas median subscapular SFT was slightly higher in the CHMS than in the US sample (7.1 mm vs. 6.0 mm). These differences may be due to heterogeneity in timing of data collection, ethnic distribution, and statistical methodology (LMS vs. GAMLSS) between the two studies. Comparison with SFT in adults is a challenge due to the scarcity of adult SFT data. Data from adults in the NHANES recruited between 1971 and 1974 shows that median subscapular SFT values in the youngest adult age category (ages 18–24 years) were moderately higher than median values at age 18 years among CHMS participants (males 11.0 vs. 9.0 mm, females 13.0 vs. 12.4 mm) (*Bowen & Custer, 1984*). Considering that the NHANES data was collected prior to the obesity epidemic, the higher SFT in the US sample is unexpected. It is possible that

**Table 3** Parameter values ($\mu, \sigma, v, \tau$) and percentiles of biceps skinfold thickness (mm) by age and sex for Canadian children and youth aged 6–19 years.

| Sex | Age (years) | $\mu$ | $\sigma$ | $v$ | $\tau$ | 3rd | 10th | 25th | 50th | 75th | 90th | 97th |
|---|---|---|---|---|---|---|---|---|---|---|---|---|
| Female | 6 | 5.0119 | 0.3284 | −0.4418 | 1.8070 | 2.89 | 3.42 | 4.08 | 5.01 | 6.28 | 7.94 | 10.40 |
| | 6.5 | 5.2250 | 0.3435 | −0.4262 | 1.9030 | 2.94 | 3.50 | 4.20 | 5.22 | 6.64 | 8.49 | 11.18 |
| | 7 | 5.4445 | 0.3591 | −0.4114 | 2.0020 | 3.00 | 3.57 | 4.32 | 5.44 | 7.03 | 9.07 | 12.01 |
| | 7.5 | 5.6746 | 0.3742 | −0.3977 | 2.1006 | 3.06 | 3.65 | 4.45 | 5.67 | 7.44 | 9.69 | 12.89 |
| | 8 | 5.9118 | 0.3874 | −0.3847 | 2.1941 | 3.13 | 3.74 | 4.58 | 5.91 | 7.85 | 10.31 | 13.76 |
| | 8.5 | 6.1425 | 0.3979 | −0.3716 | 2.2785 | 3.20 | 3.83 | 4.71 | 6.14 | 8.25 | 10.89 | 14.55 |
| | 9 | 6.3538 | 0.4052 | −0.3583 | 2.3518 | 3.27 | 3.92 | 4.84 | 6.35 | 8.60 | 11.39 | 15.20 |
| | 9.5 | 6.5358 | 0.4087 | −0.3444 | 2.4148 | 3.34 | 4.01 | 4.95 | 6.54 | 8.88 | 11.76 | 15.64 |
| | 10 | 6.6843 | 0.4085 | −0.3318 | 2.4636 | 3.42 | 4.09 | 5.06 | 6.68 | 9.09 | 12.01 | 15.87 |
| | 10.5 | 6.8017 | 0.4061 | −0.3223 | 2.4902 | 3.48 | 4.17 | 5.15 | 6.80 | 9.24 | 12.16 | 15.97 |
| | 11 | 6.8887 | 0.4035 | −0.3159 | 2.4912 | 3.54 | 4.23 | 5.22 | 6.89 | 9.33 | 12.25 | 16.03 |
| | 11.5 | 6.9462 | 0.4021 | −0.3131 | 2.4669 | 3.57 | 4.28 | 5.27 | 6.95 | 9.39 | 12.31 | 16.11 |
| | 12 | 6.9848 | 0.4016 | −0.3135 | 2.4202 | 3.59 | 4.30 | 5.31 | 6.98 | 9.43 | 12.36 | 16.22 |
| | 12.5 | 7.0172 | 0.4014 | −0.3167 | 2.3565 | 3.60 | 4.33 | 5.35 | 7.02 | 9.45 | 12.41 | 16.35 |
| | 13 | 7.0502 | 0.4005 | −0.3224 | 2.2830 | 3.62 | 4.36 | 5.39 | 7.05 | 9.46 | 12.44 | 16.48 |
| | 13.5 | 7.0857 | 0.3986 | −0.3305 | 2.2067 | 3.65 | 4.40 | 5.44 | 7.09 | 9.47 | 12.47 | 16.60 |
| | 14 | 7.1271 | 0.3956 | −0.3402 | 2.1308 | 3.69 | 4.45 | 5.49 | 7.13 | 9.48 | 12.48 | 16.69 |
| | 14.5 | 7.1747 | 0.3910 | −0.3507 | 2.0577 | 3.74 | 4.52 | 5.56 | 7.17 | 9.49 | 12.47 | 16.75 |
| | 15 | 7.2244 | 0.3855 | −0.3612 | 1.9904 | 3.79 | 4.59 | 5.64 | 7.22 | 9.49 | 12.45 | 16.76 |
| | 15.5 | 7.2719 | 0.3802 | −0.3707 | 1.9315 | 3.85 | 4.65 | 5.71 | 7.27 | 9.49 | 12.42 | 16.75 |
| | 16 | 7.3094 | 0.3760 | −0.3782 | 1.8827 | 3.89 | 4.71 | 5.76 | 7.31 | 9.49 | 12.40 | 16.75 |
| | 16.5 | 7.3323 | 0.3737 | −0.3836 | 1.8459 | 3.92 | 4.74 | 5.80 | 7.33 | 9.49 | 12.39 | 16.77 |
| | 17 | 7.3447 | 0.3733 | −0.3873 | 1.8256 | 3.93 | 4.75 | 5.82 | 7.34 | 9.49 | 12.40 | 16.82 |
| | 17.5 | 7.3491 | 0.3747 | −0.3893 | 1.8254 | 3.92 | 4.75 | 5.81 | 7.35 | 9.51 | 12.44 | 16.91 |
| | 18 | 7.3424 | 0.3774 | −0.3895 | 1.8451 | 3.90 | 4.73 | 5.80 | 7.34 | 9.53 | 12.49 | 16.99 |
| | 18.5 | 7.3213 | 0.3812 | −0.3886 | 1.8817 | 3.88 | 4.69 | 5.76 | 7.32 | 9.55 | 12.54 | 17.08 |
| | 19 | 7.2844 | 0.3863 | −0.3872 | 1.9314 | 3.83 | 4.64 | 5.70 | 7.28 | 9.56 | 12.59 | 17.16 |
| | 19.5 | 7.2355 | 0.3926 | −0.3849 | 1.9906 | 3.78 | 4.57 | 5.62 | 7.24 | 9.56 | 12.65 | 17.26 |
| Male | 6 | 4.6047 | 0.4244 | −0.4739 | 3.4627 | 2.42 | 2.79 | 3.40 | 4.60 | 6.56 | 8.87 | 11.71 |
| | 6.5 | 4.7687 | 0.4305 | −0.4661 | 3.5181 | 2.48 | 2.87 | 3.50 | 4.77 | 6.84 | 9.27 | 12.27 |
| | 7 | 4.9358 | 0.4363 | −0.4586 | 3.5741 | 2.55 | 2.95 | 3.61 | 4.94 | 7.12 | 9.69 | 12.83 |
| | 7.5 | 5.1030 | 0.4417 | −0.4516 | 3.6290 | 2.61 | 3.03 | 3.71 | 5.10 | 7.40 | 10.10 | 13.39 |
| | 8 | 5.2617 | 0.4471 | −0.4447 | 3.6814 | 2.67 | 3.10 | 3.81 | 5.26 | 7.67 | 10.50 | 13.93 |
| | 8.5 | 5.4051 | 0.4527 | −0.4387 | 3.7288 | 2.72 | 3.17 | 3.90 | 5.41 | 7.92 | 10.89 | 14.47 |
| | 9 | 5.5233 | 0.4601 | −0.4341 | 3.7636 | 2.76 | 3.21 | 3.96 | 5.52 | 8.16 | 11.26 | 15.03 |
| | 9.5 | 5.6104 | 0.4689 | −0.4318 | 3.7813 | 2.77 | 3.23 | 4.00 | 5.61 | 8.35 | 11.62 | 15.61 |
| | 10 | 5.6625 | 0.4775 | −0.4333 | 3.7815 | 2.76 | 3.23 | 4.01 | 5.66 | 8.50 | 11.92 | 16.15 |
| | 10.5 | 5.6775 | 0.4843 | −0.4396 | 3.7661 | 2.75 | 3.22 | 4.00 | 5.68 | 8.58 | 12.13 | 16.58 |
| | 11 | 5.6570 | 0.4881 | −0.4520 | 3.7424 | 2.73 | 3.20 | 3.98 | 5.66 | 8.58 | 12.21 | 16.85 |
| | 11.5 | 5.6073 | 0.4872 | −0.4719 | 3.7241 | 2.72 | 3.18 | 3.96 | 5.61 | 8.51 | 12.16 | 16.91 |

**Table 3** (*continued*)

| Sex | Age (years) | $\mu$ | $\sigma$ | $\nu$ | $\tau$ | 3rd | 10th | 25th | 50th | 75th | 90th | 97th |
|---|---|---|---|---|---|---|---|---|---|---|---|---|
| | 12 | 5.5336 | 0.4807 | −0.5000 | 3.7181 | 2.73 | 3.18 | 3.93 | 5.53 | 8.37 | 11.97 | 16.71 |
| | 12.5 | 5.4364 | 0.4702 | −0.5358 | 3.7237 | 2.73 | 3.17 | 3.89 | 5.44 | 8.17 | 11.66 | 16.31 |
| | 13 | 5.3165 | 0.4566 | −0.5778 | 3.7347 | 2.74 | 3.16 | 3.85 | 5.32 | 7.91 | 11.24 | 15.74 |
| | 13.5 | 5.1723 | 0.4417 | −0.6236 | 3.7404 | 2.74 | 3.13 | 3.79 | 5.17 | 7.62 | 10.76 | 15.05 |
| | 14 | 5.0096 | 0.4270 | −0.6718 | 3.7345 | 2.72 | 3.09 | 3.71 | 5.01 | 7.30 | 10.25 | 14.33 |
| | 14.5 | 4.8405 | 0.4130 | −0.7220 | 3.7191 | 2.69 | 3.05 | 3.63 | 4.84 | 6.98 | 9.76 | 13.64 |
| | 15 | 4.6786 | 0.4001 | −0.7736 | 3.6995 | 2.66 | 3.00 | 3.54 | 4.68 | 6.68 | 9.30 | 13.02 |
| | 15.5 | 4.5328 | 0.3883 | −0.8253 | 3.6730 | 2.63 | 2.95 | 3.46 | 4.53 | 6.41 | 8.89 | 12.49 |
| | 16 | 4.4054 | 0.3778 | −0.8752 | 3.6340 | 2.60 | 2.90 | 3.39 | 4.41 | 6.18 | 8.54 | 12.04 |
| | 16.5 | 4.2974 | 0.3687 | −0.9217 | 3.5769 | 2.58 | 2.87 | 3.34 | 4.30 | 5.98 | 8.25 | 11.68 |
| | 17 | 4.2135 | 0.3604 | −0.9645 | 3.5010 | 2.56 | 2.84 | 3.29 | 4.21 | 5.82 | 8.01 | 11.40 |
| | 17.5 | 4.1592 | 0.3521 | −1.0039 | 3.4112 | 2.56 | 2.84 | 3.28 | 4.16 | 5.70 | 7.81 | 11.17 |
| | 18 | 4.1304 | 0.3435 | −1.0395 | 3.3156 | 2.57 | 2.85 | 3.28 | 4.13 | 5.61 | 7.65 | 10.95 |
| | 18.5 | 4.1142 | 0.3347 | −1.0703 | 3.2195 | 2.59 | 2.86 | 3.29 | 4.11 | 5.53 | 7.49 | 10.70 |
| | 19 | 4.0987 | 0.3261 | −1.0957 | 3.1256 | 2.61 | 2.88 | 3.30 | 4.10 | 5.46 | 7.34 | 10.43 |

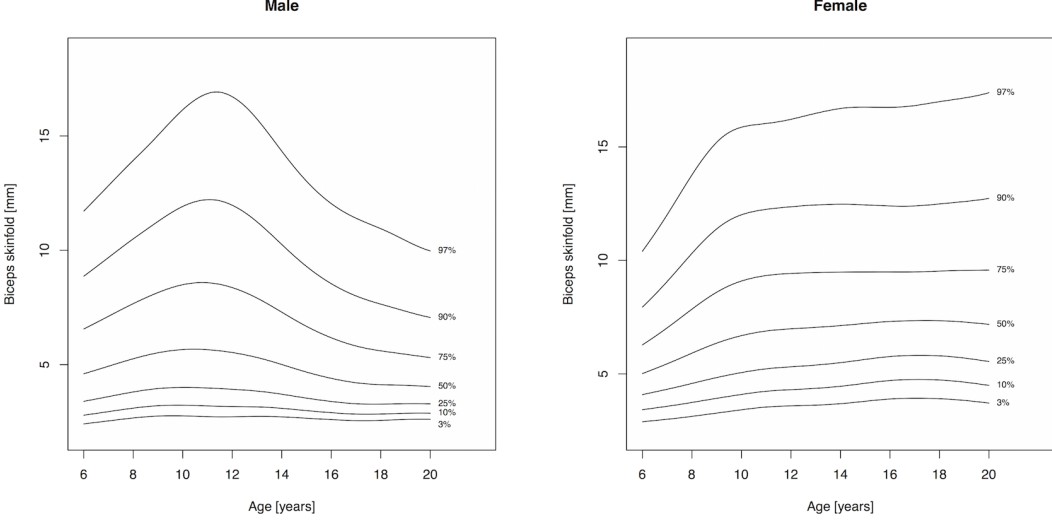

**Figure 1** Percentile curves for biceps skinfold thickness for male and female Canadian children and youth aged 6–19 years.

these differences reflect the higher rate of obesity in the US compared to Canada (*Ng et al., 2014*) or the use of a broader age category and the influence of increasing SFT in early adulthood.

Skinfold thickness measurements are frequently used to derive an estimate of body fat percentage (*Laurson, Eisenmann & Welk, 2011*; *Haas, Liepold & Schwandt, 2011*; *Rodríguez et al., 2005*). One of the most commonly used estimation equations for this purpose was developed by *Slaughter et al. (1988)* and predicts body fat from triceps and subscapular SFT. However, while the simplicity of these equations is very appealing, they are based on a historical population and their validity for use in contemporary populations is questionable

**Table 4** Parameter values ($\mu$, $\sigma$, $v$, $\tau$) and percentiles of triceps skinfold thickness (mm) by age and sex for Canadian children and youth aged 6–19 years.

| Sex | Age (years) | $\mu$ | $\sigma$ | $v$ | $\tau$ | 3rd | 10th | 25th | 50th | 75th | 90th | 97th |
|---|---|---|---|---|---|---|---|---|---|---|---|---|
| Female | 6 | 10.3527 | 0.2772 | −0.4045 | 1.7405 | 6.41 | 7.46 | 8.71 | 10.35 | 12.46 | 15.1 | 18.77 |
| | 6.5 | 10.6593 | 0.2898 | −0.3782 | 1.8402 | 6.47 | 7.55 | 8.87 | 10.66 | 12.99 | 15.85 | 19.74 |
| | 7 | 10.9767 | 0.3028 | −0.3527 | 1.9457 | 6.52 | 7.64 | 9.03 | 10.98 | 13.54 | 16.64 | 20.77 |
| | 7.5 | 11.3334 | 0.3151 | −0.3301 | 2.0578 | 6.61 | 7.75 | 9.21 | 11.33 | 14.16 | 17.50 | 21.87 |
| | 8 | 11.7320 | 0.3256 | −0.3103 | 2.1734 | 6.73 | 7.90 | 9.44 | 11.73 | 14.82 | 18.41 | 22.99 |
| | 8.5 | 12.1291 | 0.3348 | −0.2919 | 2.2869 | 6.86 | 8.06 | 9.66 | 12.13 | 15.47 | 19.29 | 24.06 |
| | 9 | 12.4911 | 0.3431 | −0.2737 | 2.3917 | 6.97 | 8.20 | 9.87 | 12.49 | 16.07 | 20.11 | 25.04 |
| | 9.5 | 12.7889 | 0.3498 | −0.2538 | 2.4833 | 7.05 | 8.31 | 10.03 | 12.79 | 16.57 | 20.77 | 25.81 |
| | 10 | 13.0158 | 0.3544 | −0.2315 | 2.5568 | 7.11 | 8.38 | 10.15 | 13.02 | 16.94 | 21.24 | 26.31 |
| | 10.5 | 13.1858 | 0.3567 | −0.2062 | 2.6095 | 7.15 | 8.44 | 10.25 | 13.19 | 17.20 | 21.54 | 26.57 |
| | 11 | 13.3113 | 0.3576 | −0.1770 | 2.6435 | 7.17 | 8.49 | 10.32 | 13.31 | 17.38 | 21.70 | 26.65 |
| | 11.5 | 13.4136 | 0.3580 | −0.1454 | 2.6597 | 7.19 | 8.52 | 10.38 | 13.41 | 17.50 | 21.80 | 26.65 |
| | 12 | 13.5272 | 0.3579 | −0.1136 | 2.6561 | 7.20 | 8.57 | 10.46 | 13.53 | 17.63 | 21.90 | 26.67 |
| | 12.5 | 13.6910 | 0.3569 | −0.0836 | 2.6324 | 7.26 | 8.66 | 10.59 | 13.69 | 17.80 | 22.05 | 26.77 |
| | 13 | 13.9211 | 0.3550 | −0.0573 | 2.5904 | 7.36 | 8.80 | 10.79 | 13.92 | 18.04 | 22.29 | 26.99 |
| | 13.5 | 14.2201 | 0.3518 | −0.0366 | 2.5346 | 7.52 | 9.02 | 11.05 | 14.22 | 18.34 | 22.60 | 27.32 |
| | 14 | 14.5871 | 0.3466 | −0.0219 | 2.4691 | 7.74 | 9.30 | 11.39 | 14.59 | 18.71 | 22.97 | 27.72 |
| | 14.5 | 14.9915 | 0.3394 | −0.0108 | 2.3963 | 8.03 | 9.65 | 11.78 | 14.99 | 19.09 | 23.34 | 28.10 |
| | 15 | 15.3946 | 0.3311 | −0.0005 | 2.3207 | 8.34 | 10.02 | 12.19 | 15.39 | 19.44 | 23.66 | 28.42 |
| | 15.5 | 15.7700 | 0.3229 | 0.0106 | 2.2476 | 8.64 | 10.37 | 12.58 | 15.77 | 19.75 | 23.93 | 28.68 |
| | 16 | 16.0934 | 0.3159 | 0.0236 | 2.1810 | 8.90 | 10.68 | 12.92 | 16.09 | 20.02 | 24.15 | 28.88 |
| | 16.5 | 16.3567 | 0.3106 | 0.0384 | 2.1230 | 9.09 | 10.93 | 13.2 | 16.36 | 20.23 | 24.33 | 29.04 |
| | 17 | 16.5739 | 0.3074 | 0.0549 | 2.0757 | 9.23 | 11.11 | 13.42 | 16.57 | 20.42 | 24.51 | 29.22 |
| | 17.5 | 16.7618 | 0.3058 | 0.0727 | 2.0406 | 9.33 | 11.25 | 13.6 | 16.76 | 20.60 | 24.69 | 29.41 |
| | 18 | 16.9205 | 0.3048 | 0.0917 | 2.0177 | 9.39 | 11.36 | 13.74 | 16.92 | 20.75 | 24.85 | 29.57 |
| | 18.5 | 17.0364 | 0.3044 | 0.1114 | 2.0040 | 9.43 | 11.43 | 13.84 | 17.04 | 20.87 | 24.96 | 29.67 |
| | 19 | 17.1018 | 0.3049 | 0.1318 | 1.9946 | 9.42 | 11.45 | 13.89 | 17.10 | 20.95 | 25.03 | 29.73 |
| | 19.5 | 17.1266 | 0.3072 | 0.1525 | 1.9862 | 9.35 | 11.41 | 13.88 | 17.13 | 20.99 | 25.10 | 29.81 |
| Male | 6 | 9.4200 | 0.3219 | −0.2987 | 2.2897 | 5.44 | 6.36 | 7.57 | 9.42 | 11.90 | 14.71 | 18.18 |
| | 6.5 | 9.6662 | 0.3372 | −0.2968 | 2.4350 | 5.47 | 6.40 | 7.66 | 9.67 | 12.41 | 15.47 | 19.20 |
| | 7 | 9.9299 | 0.3520 | −0.2951 | 2.5915 | 5.51 | 6.45 | 7.77 | 9.93 | 12.94 | 16.29 | 20.29 |
| | 7.5 | 10.2151 | 0.3651 | −0.2933 | 2.7567 | 5.57 | 6.52 | 7.89 | 10.22 | 13.51 | 17.12 | 21.37 |
| | 8 | 10.5047 | 0.3761 | −0.2908 | 2.9221 | 5.65 | 6.61 | 8.03 | 10.50 | 14.07 | 17.93 | 22.41 |
| | 8.5 | 10.7871 | 0.3856 | −0.2881 | 3.0790 | 5.73 | 6.71 | 8.16 | 10.79 | 14.60 | 18.71 | 23.38 |
| | 9 | 11.0506 | 0.3943 | −0.2861 | 3.2188 | 5.81 | 6.79 | 8.29 | 11.05 | 15.11 | 19.45 | 24.32 |
| | 9.5 | 11.2822 | 0.4020 | −0.2859 | 3.3368 | 5.88 | 6.87 | 8.41 | 11.28 | 15.56 | 20.11 | 25.18 |
| | 10 | 11.4608 | 0.4081 | −0.2879 | 3.4329 | 5.93 | 6.93 | 8.49 | 11.46 | 15.91 | 20.65 | 25.88 |
| | 10.5 | 11.5533 | 0.4127 | −0.2912 | 3.5032 | 5.95 | 6.95 | 8.53 | 11.55 | 16.13 | 20.99 | 26.35 |
| | 11 | 11.5478 | 0.4160 | −0.2954 | 3.5419 | 5.92 | 6.92 | 8.50 | 11.55 | 16.18 | 21.12 | 26.55 |
| | 11.5 | 11.4527 | 0.4182 | −0.3004 | 3.5484 | 5.86 | 6.85 | 8.42 | 11.45 | 16.08 | 21.04 | 26.51 |

**Table 4** (*continued*)

| Sex | Age (years) | $\mu$ | $\sigma$ | $\nu$ | $\tau$ | 3rd | 10th | 25th | 50th | 75th | 90th | 97th |
|-----|-------------|-------|----------|-------|--------|-----|------|------|------|------|------|------|
| | 12 | 11.2775 | 0.4193 | −0.3065 | 3.5262 | 5.77 | 6.74 | 8.29 | 11.28 | 15.85 | 20.77 | 26.25 |
| | 12.5 | 11.0300 | 0.4197 | −0.3135 | 3.4821 | 5.64 | 6.60 | 8.11 | 11.03 | 15.50 | 20.35 | 25.79 |
| | 13 | 10.7229 | 0.4191 | −0.3213 | 3.4194 | 5.49 | 6.43 | 7.90 | 10.72 | 15.05 | 19.79 | 25.16 |
| | 13.5 | 10.3697 | 0.4183 | −0.3293 | 3.3374 | 5.32 | 6.23 | 7.65 | 10.37 | 14.53 | 19.13 | 24.42 |
| | 14 | 9.9979 | 0.4173 | −0.3373 | 3.2383 | 5.14 | 6.02 | 7.40 | 10.00 | 13.98 | 18.43 | 23.64 |
| | 14.5 | 9.6331 | 0.4169 | −0.3444 | 3.1272 | 4.95 | 5.82 | 7.15 | 9.63 | 13.44 | 17.75 | 22.89 |
| | 15 | 9.3037 | 0.4173 | −0.3498 | 3.0095 | 4.77 | 5.62 | 6.92 | 9.30 | 12.95 | 17.15 | 22.27 |
| | 15.5 | 9.0358 | 0.4179 | −0.3528 | 2.8902 | 4.62 | 5.47 | 6.73 | 9.04 | 12.55 | 16.66 | 21.78 |
| | 16 | 8.8355 | 0.4189 | −0.3524 | 2.7730 | 4.50 | 5.34 | 6.59 | 8.84 | 12.24 | 16.28 | 21.44 |
| | 16.5 | 8.6954 | 0.4204 | −0.3477 | 2.6613 | 4.40 | 5.25 | 6.50 | 8.70 | 12.02 | 16.02 | 21.23 |
| | 17 | 8.6080 | 0.4222 | −0.3389 | 2.5605 | 4.33 | 5.19 | 6.44 | 8.61 | 11.88 | 15.86 | 21.12 |
| | 17.5 | 8.5657 | 0.4241 | −0.3264 | 2.4764 | 4.28 | 5.15 | 6.41 | 8.57 | 11.80 | 15.76 | 21.07 |
| | 18 | 8.5609 | 0.4257 | −0.3112 | 2.4147 | 4.24 | 5.13 | 6.41 | 8.56 | 11.77 | 15.73 | 21.05 |
| | 18.5 | 8.5809 | 0.4262 | −0.2939 | 2.3794 | 4.23 | 5.14 | 6.42 | 8.58 | 11.78 | 15.71 | 21.00 |
| | 19 | 8.6121 | 0.4256 | −0.2749 | 2.3676 | 4.23 | 5.15 | 6.45 | 8.61 | 11.80 | 15.69 | 20.89 |

**Table 5** Parameter values ($\mu$, $\sigma$, $\nu$, $\tau$) and percentiles of subscapular skinfold thickness (mm) by age and sex for Canadian children and youth aged 6–19 years.

| Sex | Age (years) | $\mu$ | $\sigma$ | $\nu$ | $\tau$ | 3rd | 10th | 25th | 50th | 75th | 90th | 97th |
|-----|-------------|-------|----------|-------|--------|-----|------|------|------|------|------|------|
| Female | 6 | 5.7916 | 0.3158 | −1.4955 | 2.0865 | 3.79 | 4.21 | 4.79 | 5.75 | 7.38 | 10.19 | 16.69 |
| | 6.5 | 5.9375 | 0.3313 | −1.4087 | 2.2978 | 3.81 | 4.24 | 4.84 | 5.90 | 7.75 | 10.90 | 18.18 |
| | 7 | 6.0924 | 0.3474 | −1.3221 | 2.5290 | 3.83 | 4.26 | 4.90 | 6.06 | 8.15 | 11.67 | 19.75 |
| | 7.5 | 6.2641 | 0.3640 | −1.2361 | 2.7759 | 3.86 | 4.30 | 4.96 | 6.24 | 8.58 | 12.51 | 21.34 |
| | 8 | 6.4656 | 0.3805 | −1.1513 | 3.0269 | 3.90 | 4.35 | 5.05 | 6.45 | 9.07 | 13.41 | 22.83 |
| | 8.5 | 6.6905 | 0.3963 | −1.0679 | 3.2661 | 3.95 | 4.41 | 5.15 | 6.68 | 9.59 | 14.29 | 24.02 |
| | 9 | 6.9464 | 0.4105 | −0.9870 | 3.4737 | 4.01 | 4.49 | 5.28 | 6.94 | 10.12 | 15.12 | 24.80 |
| | 9.5 | 7.2332 | 0.4223 | −0.9091 | 3.6298 | 4.09 | 4.59 | 5.43 | 7.23 | 10.66 | 15.86 | 25.18 |
| | 10 | 7.5377 | 0.4317 | −0.8357 | 3.7182 | 4.18 | 4.71 | 5.61 | 7.54 | 11.18 | 16.49 | 25.34 |
| | 10.5 | 7.8500 | 0.4388 | −0.7676 | 3.7348 | 4.27 | 4.84 | 5.80 | 7.85 | 11.66 | 17.05 | 25.46 |
| | 11 | 8.1575 | 0.4441 | −0.7052 | 3.6890 | 4.37 | 4.98 | 5.99 | 8.16 | 12.11 | 17.53 | 25.60 |
| | 11.5 | 8.4551 | 0.4477 | −0.6491 | 3.5998 | 4.46 | 5.11 | 6.19 | 8.46 | 12.52 | 17.95 | 25.77 |
| | 12 | 8.7644 | 0.4495 | −0.6001 | 3.4892 | 4.56 | 5.26 | 6.40 | 8.76 | 12.92 | 18.35 | 26.00 |
| | 12.5 | 9.1037 | 0.4494 | −0.5588 | 3.3729 | 4.70 | 5.44 | 6.65 | 9.10 | 13.34 | 18.79 | 26.34 |
| | 13 | 9.4771 | 0.4476 | −0.5258 | 3.2654 | 4.86 | 5.66 | 6.93 | 9.48 | 13.79 | 19.28 | 26.77 |
| | 13.5 | 9.8715 | 0.4441 | −0.5003 | 3.1753 | 5.05 | 5.90 | 7.24 | 9.87 | 14.26 | 19.79 | 27.26 |
| | 14 | 10.2792 | 0.4393 | −0.4810 | 3.1074 | 5.26 | 6.16 | 7.56 | 10.28 | 14.74 | 20.30 | 27.74 |
| | 14.5 | 10.6873 | 0.4334 | −0.4669 | 3.0654 | 5.50 | 6.44 | 7.89 | 10.69 | 15.22 | 20.80 | 28.18 |
| | 15 | 11.0716 | 0.4272 | −0.4570 | 3.0425 | 5.73 | 6.70 | 8.21 | 11.07 | 15.66 | 21.25 | 28.56 |
| | 15.5 | 11.4057 | 0.4219 | −0.4497 | 3.0273 | 5.93 | 6.94 | 8.49 | 11.41 | 16.04 | 21.65 | 28.91 |
| | 16 | 11.6741 | 0.4182 | −0.4439 | 3.0112 | 6.09 | 7.13 | 8.71 | 11.67 | 16.36 | 21.98 | 29.23 |
| | 16.5 | 11.8843 | 0.4163 | −0.4389 | 2.9901 | 6.21 | 7.27 | 8.88 | 11.88 | 16.61 | 22.27 | 29.55 |

**Table 5** (*continued*)

| Sex | Age (years) | μ | σ | ν | τ | 3rd | 10th | 25th | 50th | 75th | 90th | 97th |
|------|------|------|------|------|------|------|------|------|------|------|------|------|
| | 17 | 12.0601 | 0.4156 | −0.4342 | 2.9646 | 6.30 | 7.38 | 9.01 | 12.06 | 16.83 | 22.54 | 29.90 |
| | 17.5 | 12.2198 | 0.4158 | −0.4289 | 2.9386 | 6.37 | 7.47 | 9.14 | 12.22 | 17.04 | 22.81 | 30.26 |
| | 18 | 12.3746 | 0.4158 | −0.4219 | 2.9175 | 6.44 | 7.56 | 9.25 | 12.37 | 17.24 | 23.06 | 30.56 |
| | 18.5 | 12.5249 | 0.4154 | −0.4125 | 2.9008 | 6.51 | 7.65 | 9.37 | 12.52 | 17.42 | 23.28 | 30.79 |
| | 19 | 12.6650 | 0.4149 | −0.4006 | 2.8872 | 6.57 | 7.73 | 9.47 | 12.67 | 17.59 | 23.45 | 30.94 |
| | 19.5 | 12.7914 | 0.4146 | −0.3862 | 2.8775 | 6.62 | 7.80 | 9.57 | 12.79 | 17.75 | 23.60 | 31.04 |
| Male | 6 | 5.2871 | 0.3836 | −1.0960 | 4.8016 | 3.21 | 3.51 | 4.05 | 5.29 | 7.70 | 11.28 | 17.20 |
| | 6.5 | 5.5388 | 0.3904 | −1.0977 | 5.0401 | 3.35 | 3.66 | 4.22 | 5.54 | 8.16 | 12.09 | 18.69 |
| | 7 | 5.8079 | 0.3965 | −1.0994 | 5.2828 | 3.49 | 3.82 | 4.41 | 5.81 | 8.64 | 12.96 | 20.26 |
| | 7.5 | 6.0797 | 0.4021 | −1.1001 | 5.5045 | 3.64 | 3.97 | 4.59 | 6.08 | 9.13 | 13.84 | 21.88 |
| | 8 | 6.3370 | 0.4077 | −1.0994 | 5.6729 | 3.78 | 4.12 | 4.77 | 6.33 | 9.60 | 14.71 | 23.55 |
| | 8.5 | 6.5751 | 0.4133 | −1.0974 | 5.7598 | 3.90 | 4.25 | 4.93 | 6.57 | 10.03 | 15.54 | 25.25 |
| | 9 | 6.7747 | 0.4203 | −1.0946 | 5.7440 | 3.99 | 4.36 | 5.06 | 6.77 | 10.42 | 16.36 | 27.21 |
| | 9.5 | 6.9283 | 0.4281 | −1.0916 | 5.6129 | 4.05 | 4.43 | 5.15 | 6.92 | 10.74 | 17.12 | 29.36 |
| | 10 | 7.0323 | 0.4352 | −1.0889 | 5.3693 | 4.07 | 4.47 | 5.21 | 7.02 | 10.96 | 17.69 | 31.32 |
| | 10.5 | 7.0859 | 0.4402 | −1.0870 | 5.0319 | 4.07 | 4.48 | 5.24 | 7.07 | 11.04 | 17.99 | 32.76 |
| | 11 | 7.1096 | 0.4423 | −1.0872 | 4.6343 | 4.07 | 4.49 | 5.26 | 7.08 | 11.03 | 18.00 | 33.47 |
| | 11.5 | 7.1329 | 0.4406 | −1.0906 | 4.2149 | 4.08 | 4.52 | 5.30 | 7.10 | 10.94 | 17.78 | 33.43 |
| | 12 | 7.1737 | 0.4348 | −1.0971 | 3.8024 | 4.11 | 4.58 | 5.37 | 7.14 | 10.83 | 17.39 | 32.71 |
| | 12.5 | 7.2268 | 0.4254 | −1.1057 | 3.4165 | 4.17 | 4.66 | 5.46 | 7.19 | 10.69 | 16.88 | 31.45 |
| | 13 | 7.2915 | 0.4130 | −1.1155 | 3.0718 | 4.24 | 4.76 | 5.57 | 7.26 | 10.54 | 16.30 | 29.85 |
| | 13.5 | 7.3647 | 0.3986 | −1.1256 | 2.7784 | 4.33 | 4.88 | 5.70 | 7.33 | 10.40 | 15.71 | 28.13 |
| | 14 | 7.4489 | 0.3831 | −1.1352 | 2.5417 | 4.44 | 5.01 | 5.85 | 7.42 | 10.28 | 15.16 | 26.46 |
| | 14.5 | 7.5468 | 0.3677 | −1.1438 | 2.3642 | 4.56 | 5.15 | 6.00 | 7.52 | 10.20 | 14.71 | 24.97 |
| | 15 | 7.6669 | 0.3533 | −1.1508 | 2.2436 | 4.70 | 5.31 | 6.16 | 7.65 | 10.18 | 14.39 | 23.76 |
| | 15.5 | 7.8138 | 0.3410 | −1.1553 | 2.1705 | 4.85 | 5.47 | 6.34 | 7.80 | 10.24 | 14.23 | 22.92 |
| | 16 | 7.9923 | 0.3316 | −1.1565 | 2.1345 | 5.01 | 5.64 | 6.52 | 7.98 | 10.37 | 14.24 | 22.45 |
| | 16.5 | 8.2078 | 0.3247 | −1.1538 | 2.1262 | 5.18 | 5.83 | 6.73 | 8.20 | 10.59 | 14.39 | 22.30 |
| | 17 | 8.4528 | 0.3197 | −1.1467 | 2.1359 | 5.36 | 6.03 | 6.94 | 8.44 | 10.86 | 14.65 | 22.35 |
| | 17.5 | 8.7139 | 0.3161 | −1.1350 | 2.1547 | 5.55 | 6.23 | 7.17 | 8.71 | 11.17 | 14.98 | 22.53 |
| | 18 | 8.9755 | 0.3145 | −1.1183 | 2.1759 | 5.72 | 6.42 | 7.38 | 8.97 | 11.50 | 15.35 | 22.83 |
| | 18.5 | 9.2212 | 0.3145 | −1.0964 | 2.1938 | 5.87 | 6.59 | 7.58 | 9.22 | 11.82 | 15.73 | 23.20 |
| | 19 | 9.4349 | 0.3157 | −1.0691 | 2.2048 | 5.98 | 6.72 | 7.75 | 9.43 | 12.10 | 16.08 | 23.54 |

(*Wells & Fewtrell, 2006*) as evidenced by the bias when compared with methods like dual-energy X-ray absorptiometry (*Rodríguez et al., 2005*; *Wells et al., 1999*; *Freedman, Horlick & Berenson, 2013*). Moreover, reference data for directly measured body fat using criterion methods are now available (*Moreno et al., 2007*; *Wells et al., 2012*; *Van der Sluis et al., 2002*) that allow for accurate assessment of the development of lean and fat mass in children. It should also be acknowledged in this context that SFT inherently only measures external fat and can not assess internal visceral adiposity, which is most strongly associated with health outcomes (*Kelishadi et al., 2015*). However, SFT shows good correlations with elevated levels of cardiovascular disease risk factors (*Steinberger et al., 2005*; *Petkeviciene et al., 2015*) and metabolic syndrome (*Laurson, Eisenmann & Welk, 2011*).

**Table 6** Parameter values ($\mu$, $\sigma$, $v$, $\tau$) and percentiles of suprailiac skinfold thickness (mm) by age and sex for Canadian children and youth aged 6–19 years.

| Sex | Age (years) | $\mu$ | $\sigma$ | $v$ | $\tau$ | 3rd | 10th | 25th | 50th | 75th | 90th | 97th |
|---|---|---|---|---|---|---|---|---|---|---|---|---|
| Female | 6 | 6.6972 | 0.4008 | −0.9029 | 3.3367 | 3.86 | 4.34 | 5.11 | 6.70 | 9.58 | 13.80 | 21.08 |
| | 6.5 | 6.8654 | 0.4266 | −0.8326 | 3.4717 | 3.81 | 4.32 | 5.14 | 6.86 | 10.07 | 14.77 | 22.74 |
| | 7 | 7.0730 | 0.4535 | −0.7626 | 3.6134 | 3.78 | 4.31 | 5.18 | 7.07 | 10.64 | 15.87 | 24.51 |
| | 7.5 | 7.3765 | 0.4800 | −0.6931 | 3.7620 | 3.79 | 4.34 | 5.29 | 7.38 | 11.37 | 17.17 | 26.42 |
| | 8 | 7.8030 | 0.5037 | −0.6240 | 3.9130 | 3.86 | 4.45 | 5.48 | 7.80 | 12.28 | 18.65 | 28.33 |
| | 8.5 | 8.2923 | 0.5230 | −0.5551 | 4.0595 | 3.95 | 4.59 | 5.72 | 8.29 | 13.24 | 20.07 | 29.86 |
| | 9 | 8.8705 | 0.5363 | −0.4870 | 4.1943 | 4.10 | 4.80 | 6.03 | 8.87 | 14.27 | 21.43 | 31.02 |
| | 9.5 | 9.5357 | 0.5422 | −0.4201 | 4.3053 | 4.30 | 5.06 | 6.42 | 9.54 | 15.34 | 22.66 | 31.82 |
| | 10 | 10.2176 | 0.5420 | −0.3544 | 4.3718 | 4.53 | 5.36 | 6.84 | 10.22 | 16.32 | 23.65 | 32.26 |
| | 10.5 | 10.9024 | 0.5379 | −0.2901 | 4.3767 | 4.76 | 5.67 | 7.28 | 10.90 | 17.22 | 24.46 | 32.54 |
| | 11 | 11.5991 | 0.5315 | −0.2274 | 4.3136 | 5.00 | 6.00 | 7.75 | 11.60 | 18.08 | 25.21 | 32.85 |
| | 11.5 | 12.2912 | 0.5237 | −0.1671 | 4.1882 | 5.24 | 6.34 | 8.23 | 12.29 | 18.89 | 25.89 | 33.20 |
| | 12 | 12.9890 | 0.5145 | −0.1106 | 4.0133 | 5.49 | 6.70 | 8.74 | 12.99 | 19.65 | 26.54 | 33.62 |
| | 12.5 | 13.7280 | 0.5043 | −0.0598 | 3.8054 | 5.77 | 7.10 | 9.30 | 13.73 | 20.45 | 27.26 | 34.22 |
| | 13 | 14.5051 | 0.4933 | −0.0163 | 3.5869 | 6.09 | 7.55 | 9.92 | 14.51 | 21.27 | 28.05 | 34.99 |
| | 13.5 | 15.2862 | 0.4817 | 0.0199 | 3.3771 | 6.43 | 8.04 | 10.56 | 15.29 | 22.07 | 28.84 | 35.81 |
| | 14 | 16.0317 | 0.4695 | 0.0499 | 3.1880 | 6.79 | 8.53 | 11.2 | 16.03 | 22.80 | 29.55 | 36.55 |
| | 14.5 | 16.7126 | 0.4571 | 0.0748 | 3.0276 | 7.14 | 9.02 | 11.81 | 16.71 | 23.44 | 30.15 | 37.16 |
| | 15 | 17.3039 | 0.4453 | 0.0953 | 2.8954 | 7.48 | 9.47 | 12.37 | 17.30 | 23.96 | 30.61 | 37.61 |
| | 15.5 | 17.7850 | 0.4348 | 0.1126 | 2.7869 | 7.78 | 9.86 | 12.84 | 17.78 | 24.36 | 30.93 | 37.90 |
| | 16 | 18.1484 | 0.4259 | 0.1274 | 2.6967 | 8.01 | 10.17 | 13.21 | 18.15 | 24.63 | 31.12 | 38.05 |
| | 16.5 | 18.4145 | 0.4185 | 0.1401 | 2.6189 | 8.20 | 10.42 | 13.50 | 18.41 | 24.80 | 31.21 | 38.09 |
| | 17 | 18.6318 | 0.4121 | 0.1503 | 2.5499 | 8.36 | 10.63 | 13.75 | 18.63 | 24.92 | 31.27 | 38.11 |
| | 17.5 | 18.8355 | 0.4064 | 0.1579 | 2.4865 | 8.51 | 10.83 | 13.98 | 18.84 | 25.04 | 31.34 | 38.16 |
| | 18 | 19.0186 | 0.4015 | 0.1640 | 2.4234 | 8.64 | 11.01 | 14.19 | 19.02 | 25.15 | 31.40 | 38.23 |
| | 18.5 | 19.1639 | 0.3973 | 0.1694 | 2.3582 | 8.75 | 11.17 | 14.37 | 19.16 | 25.22 | 31.43 | 38.29 |
| | 19 | 19.2661 | 0.3942 | 0.1752 | 2.2932 | 8.81 | 11.28 | 14.51 | 19.27 | 25.24 | 31.43 | 38.32 |
| | 19.5 | 19.3275 | 0.3924 | 0.1820 | 2.2315 | 8.84 | 11.35 | 14.60 | 19.33 | 25.24 | 31.41 | 38.34 |
| Male | 6 | 6.3018 | 0.5203 | −0.6962 | 6.1066 | 3.16 | 3.56 | 4.33 | 6.30 | 10.49 | 16.56 | 24.91 |
| | 6.5 | 6.5930 | 0.5288 | −0.6668 | 6.2092 | 3.26 | 3.67 | 4.49 | 6.59 | 11.05 | 17.42 | 25.96 |
| | 7 | 6.8933 | 0.5374 | −0.6381 | 6.3102 | 3.36 | 3.79 | 4.66 | 6.89 | 11.62 | 18.31 | 27.04 |
| | 7.5 | 7.2022 | 0.5462 | −0.6105 | 6.4014 | 3.46 | 3.91 | 4.83 | 7.20 | 12.22 | 19.25 | 28.21 |
| | 8 | 7.5126 | 0.5562 | −0.5844 | 6.4749 | 3.55 | 4.03 | 5.00 | 7.51 | 12.85 | 20.26 | 29.55 |
| | 8.5 | 7.8236 | 0.5672 | −0.5608 | 6.5276 | 3.63 | 4.13 | 5.15 | 7.82 | 13.50 | 21.36 | 31.09 |
| | 9 | 8.1293 | 0.5793 | −0.5399 | 6.5518 | 3.70 | 4.23 | 5.30 | 8.13 | 14.17 | 22.55 | 32.87 |
| | 9.5 | 8.4238 | 0.5915 | −0.5219 | 6.5395 | 3.76 | 4.31 | 5.44 | 8.42 | 14.84 | 23.76 | 34.76 |
| | 10 | 8.7035 | 0.6017 | −0.5068 | 6.4882 | 3.83 | 4.40 | 5.58 | 8.70 | 15.46 | 24.88 | 36.53 |
| | 10.5 | 8.9678 | 0.6086 | −0.4954 | 6.4021 | 3.90 | 4.49 | 5.72 | 8.97 | 16.01 | 25.84 | 38.04 |
| | 11 | 9.2171 | 0.6113 | −0.4875 | 6.2895 | 3.98 | 4.60 | 5.87 | 9.22 | 16.46 | 26.57 | 39.13 |
| | 11.5 | 9.4540 | 0.6099 | −0.4830 | 6.1530 | 4.08 | 4.72 | 6.03 | 9.45 | 16.82 | 27.06 | 39.82 |
| | 12 | 9.6782 | 0.6052 | −0.4808 | 5.9886 | 4.19 | 4.86 | 6.19 | 9.68 | 17.10 | 27.35 | 40.15 |

**Table 6** (*continued*)

| Sex | Age (years) | μ | σ | ν | τ | 3rd | 10th | 25th | 50th | 75th | 90th | 97th |
|-----|-------------|---|---|---|---|-----|------|------|------|------|------|------|
| | 12.5 | 9.8876 | 0.5978 | −0.4800 | 5.7993 | 4.31 | 5.00 | 6.36 | 9.89 | 17.30 | 27.44 | 40.13 |
| | 13 | 10.0850 | 0.5877 | −0.4799 | 5.5886 | 4.44 | 5.15 | 6.54 | 10.09 | 17.42 | 27.35 | 39.77 |
| | 13.5 | 10.2667 | 0.5761 | −0.4790 | 5.3571 | 4.57 | 5.31 | 6.72 | 10.27 | 17.47 | 27.10 | 39.16 |
| | 14 | 10.4305 | 0.5638 | −0.4759 | 5.1099 | 4.70 | 5.46 | 6.90 | 10.43 | 17.46 | 26.73 | 38.33 |
| | 14.5 | 10.5803 | 0.5516 | −0.4696 | 4.8578 | 4.81 | 5.60 | 7.06 | 10.58 | 17.42 | 26.31 | 37.43 |
| | 15 | 10.7272 | 0.5403 | −0.4599 | 4.6105 | 4.92 | 5.74 | 7.23 | 10.73 | 17.39 | 25.91 | 36.56 |
| | 15.5 | 10.8808 | 0.5305 | −0.4468 | 4.3691 | 5.02 | 5.87 | 7.39 | 10.88 | 17.37 | 25.58 | 35.84 |
| | 16 | 11.0471 | 0.5234 | −0.4305 | 4.1337 | 5.10 | 5.99 | 7.55 | 11.05 | 17.42 | 25.40 | 35.41 |
| | 16.5 | 11.2298 | 0.5191 | −0.4113 | 3.9077 | 5.17 | 6.11 | 7.71 | 11.23 | 17.53 | 25.37 | 35.28 |
| | 17 | 11.4314 | 0.5170 | −0.3898 | 3.6947 | 5.23 | 6.21 | 7.87 | 11.43 | 17.69 | 25.47 | 35.38 |
| | 17.5 | 11.6547 | 0.5158 | −0.3666 | 3.4997 | 5.29 | 6.32 | 8.04 | 11.65 | 17.90 | 25.65 | 35.59 |
| | 18 | 11.8974 | 0.5153 | −0.3423 | 3.3301 | 5.35 | 6.44 | 8.23 | 11.90 | 18.15 | 25.90 | 35.90 |
| | 18.5 | 12.1575 | 0.5147 | −0.3177 | 3.1943 | 5.41 | 6.56 | 8.42 | 12.16 | 18.43 | 26.19 | 36.21 |
| | 19 | 12.4347 | 0.5131 | −0.2935 | 3.0962 | 5.50 | 6.70 | 8.62 | 12.43 | 18.74 | 26.47 | 36.45 |

**Table 7** Parameter values (μ, σ, ν, τ) and percentiles of medial calf skinfold thickness (mm) by age and sex for Canadian children and youth aged 6–19 years.

| Sex | Age (years) | μ | σ | ν | τ | 3rd | 10th | 25th | 50th | 75th | 90th | 97th |
|-----|-------------|---|---|---|---|-----|------|------|------|------|------|------|
| Female | 6 | 8.8756 | 0.3546 | −0.1545 | 1.7494 | 4.67 | 5.76 | 7.09 | 8.88 | 11.20 | 14.12 | 18.13 |
| | 6.5 | 9.2641 | 0.3628 | −0.1300 | 1.8386 | 4.79 | 5.92 | 7.33 | 9.26 | 11.80 | 14.90 | 19.06 |
| | 7 | 9.6544 | 0.3713 | −0.1055 | 1.9318 | 4.91 | 6.08 | 7.56 | 9.65 | 12.41 | 15.70 | 20.00 |
| | 7.5 | 10.0464 | 0.3798 | −0.0817 | 2.0269 | 5.02 | 6.23 | 7.79 | 10.05 | 13.03 | 16.52 | 20.95 |
| | 8 | 10.4394 | 0.3880 | −0.0590 | 2.1171 | 5.13 | 6.38 | 8.02 | 10.44 | 13.65 | 17.34 | 21.91 |
| | 8.5 | 10.8320 | 0.3960 | −0.0376 | 2.1960 | 5.23 | 6.53 | 8.24 | 10.83 | 14.28 | 18.16 | 22.88 |
| | 9 | 11.2226 | 0.4034 | −0.0180 | 2.2604 | 5.33 | 6.67 | 8.47 | 11.22 | 14.89 | 18.97 | 23.85 |
| | 9.5 | 11.6082 | 0.4099 | −0.0009 | 2.3108 | 5.43 | 6.82 | 8.70 | 11.61 | 15.49 | 19.76 | 24.81 |
| | 10 | 11.9874 | 0.4154 | 0.0128 | 2.3480 | 5.54 | 6.98 | 8.93 | 11.99 | 16.07 | 20.52 | 25.75 |
| | 10.5 | 12.3598 | 0.4200 | 0.0225 | 2.3743 | 5.65 | 7.14 | 9.17 | 12.36 | 16.63 | 21.27 | 26.67 |
| | 11 | 12.7243 | 0.4237 | 0.0285 | 2.3936 | 5.77 | 7.30 | 9.41 | 12.72 | 17.17 | 21.99 | 27.57 |
| | 11.5 | 13.0793 | 0.4265 | 0.0308 | 2.4083 | 5.90 | 7.47 | 9.64 | 13.08 | 17.69 | 22.68 | 28.45 |
| | 12 | 13.4235 | 0.4281 | 0.0299 | 2.4180 | 6.04 | 7.65 | 9.88 | 13.42 | 18.18 | 23.33 | 29.28 |
| | 12.5 | 13.7571 | 0.4283 | 0.0271 | 2.4199 | 6.19 | 7.84 | 10.13 | 13.76 | 18.64 | 23.93 | 30.05 |
| | 13 | 14.0787 | 0.4274 | 0.0232 | 2.4129 | 6.36 | 8.04 | 10.38 | 14.08 | 19.06 | 24.47 | 30.74 |
| | 13.5 | 14.3853 | 0.4249 | 0.0189 | 2.3988 | 6.53 | 8.25 | 10.63 | 14.39 | 19.44 | 24.93 | 31.33 |
| | 14 | 14.6745 | 0.4210 | 0.0150 | 2.3798 | 6.71 | 8.47 | 10.88 | 14.67 | 19.77 | 25.31 | 31.78 |
| | 14.5 | 14.9414 | 0.4156 | 0.0139 | 2.3578 | 6.90 | 8.69 | 11.13 | 14.94 | 20.04 | 25.59 | 32.08 |
| | 15 | 15.1812 | 0.4092 | 0.0180 | 2.3359 | 7.08 | 8.90 | 11.37 | 15.18 | 20.25 | 25.76 | 32.20 |
| | 15.5 | 15.3921 | 0.4024 | 0.0278 | 2.3153 | 7.25 | 9.10 | 11.58 | 15.39 | 20.41 | 25.85 | 32.19 |
| | 16 | 15.5739 | 0.3958 | 0.0429 | 2.2962 | 7.39 | 9.27 | 11.77 | 15.57 | 20.53 | 25.88 | 32.07 |
| | 16.5 | 15.7273 | 0.3898 | 0.0617 | 2.2790 | 7.51 | 9.41 | 11.94 | 15.73 | 20.62 | 25.87 | 31.91 |
| | 17 | 15.8549 | 0.3849 | 0.0820 | 2.2666 | 7.59 | 9.53 | 12.07 | 15.85 | 20.7 | 25.84 | 31.74 |

**Table 7** (*continued*)

| Sex | Age (years) | μ | σ | ν | τ | 3rd | 10th | 25th | 50th | 75th | 90th | 97th |
|-----|-------------|-----|-----|-----|-----|-----|------|------|------|------|------|------|
| | 17.5 | 15.9602 | 0.3811 | 0.1014 | 2.2608 | 7.66 | 9.62 | 12.18 | 15.96 | 20.76 | 25.83 | 31.61 |
| | 18 | 16.0453 | 0.3785 | 0.1179 | 2.2603 | 7.70 | 9.68 | 12.26 | 16.05 | 20.82 | 25.84 | 31.52 |
| | 18.5 | 16.1127 | 0.3774 | 0.1309 | 2.2605 | 7.73 | 9.72 | 12.32 | 16.11 | 20.89 | 25.88 | 31.50 |
| | 19 | 16.1645 | 0.3779 | 0.1405 | 2.2579 | 7.72 | 9.74 | 12.35 | 16.16 | 20.95 | 25.95 | 31.57 |
| | 19.5 | 16.2024 | 0.3800 | 0.1470 | 2.2522 | 7.69 | 9.72 | 12.36 | 16.20 | 21.02 | 26.06 | 31.71 |
| Male | 6 | 7.6139 | 0.3532 | −0.4290 | 2.0886 | 4.25 | 5.03 | 6.05 | 7.61 | 9.82 | 12.62 | 16.57 |
| | 6.5 | 7.9514 | 0.3740 | −0.4000 | 2.2278 | 4.30 | 5.11 | 6.21 | 7.95 | 10.47 | 13.61 | 17.96 |
| | 7 | 8.3241 | 0.3942 | −0.3711 | 2.3743 | 4.37 | 5.21 | 6.38 | 8.32 | 11.18 | 14.70 | 19.45 |
| | 7.5 | 8.7435 | 0.4118 | −0.3426 | 2.5227 | 4.47 | 5.34 | 6.59 | 8.74 | 11.95 | 15.84 | 20.96 |
| | 8 | 9.1952 | 0.4264 | −0.3147 | 2.6629 | 4.58 | 5.50 | 6.84 | 9.20 | 12.75 | 17.00 | 22.45 |
| | 8.5 | 9.6377 | 0.4404 | −0.2878 | 2.7854 | 4.69 | 5.64 | 7.07 | 9.64 | 13.55 | 18.14 | 23.92 |
| | 9 | 10.0372 | 0.4568 | −0.2627 | 2.8827 | 4.75 | 5.74 | 7.25 | 10.04 | 14.31 | 19.31 | 25.50 |
| | 9.5 | 10.3692 | 0.4757 | −0.2405 | 2.9490 | 4.75 | 5.78 | 7.38 | 10.37 | 15.02 | 20.46 | 27.15 |
| | 10 | 10.6239 | 0.4941 | −0.2228 | 2.9851 | 4.71 | 5.78 | 7.45 | 10.62 | 15.62 | 21.48 | 28.69 |
| | 10.5 | 10.7952 | 0.5074 | −0.2113 | 2.9966 | 4.68 | 5.77 | 7.49 | 10.80 | 16.03 | 22.21 | 29.84 |
| | 11 | 10.9036 | 0.5136 | −0.2077 | 2.9961 | 4.67 | 5.79 | 7.53 | 10.90 | 16.27 | 22.62 | 30.50 |
| | 11.5 | 10.9717 | 0.5145 | −0.2128 | 2.9980 | 4.71 | 5.82 | 7.58 | 10.97 | 16.39 | 22.83 | 30.84 |
| | 12 | 11.0043 | 0.5115 | −0.2262 | 3.0093 | 4.76 | 5.87 | 7.62 | 11.00 | 16.43 | 22.88 | 30.95 |
| | 12.5 | 10.9698 | 0.5073 | −0.2460 | 3.0302 | 4.81 | 5.90 | 7.63 | 10.97 | 16.35 | 22.80 | 30.88 |
| | 13 | 10.8526 | 0.5017 | −0.2700 | 3.0602 | 4.84 | 5.90 | 7.58 | 10.85 | 16.15 | 22.51 | 30.54 |
| | 13.5 | 10.6446 | 0.4947 | −0.2959 | 3.0972 | 4.83 | 5.86 | 7.48 | 10.64 | 15.79 | 22.00 | 29.86 |
| | 14 | 10.3584 | 0.4868 | −0.3214 | 3.1372 | 4.80 | 5.77 | 7.32 | 10.36 | 15.31 | 21.30 | 28.89 |
| | 14.5 | 10.0042 | 0.4798 | −0.3446 | 3.1788 | 4.71 | 5.64 | 7.11 | 10.00 | 14.75 | 20.47 | 27.75 |
| | 15 | 9.6173 | 0.4750 | −0.3643 | 3.2226 | 4.58 | 5.46 | 6.85 | 9.62 | 14.15 | 19.64 | 26.61 |
| | 15.5 | 9.2415 | 0.4714 | −0.3799 | 3.2679 | 4.45 | 5.28 | 6.60 | 9.24 | 13.59 | 18.85 | 25.52 |
| | 16 | 8.9001 | 0.4694 | −0.3911 | 3.3148 | 4.31 | 5.10 | 6.37 | 8.90 | 13.09 | 18.15 | 24.57 |
| | 16.5 | 8.5901 | 0.4703 | −0.3978 | 3.3649 | 4.17 | 4.92 | 6.14 | 8.59 | 12.66 | 17.59 | 23.82 |
| | 17 | 8.3241 | 0.4727 | −0.4010 | 3.4191 | 4.03 | 4.76 | 5.93 | 8.32 | 12.32 | 17.15 | 23.24 |
| | 17.5 | 8.1317 | 0.4742 | −0.4017 | 3.4752 | 3.93 | 4.64 | 5.79 | 8.13 | 12.06 | 16.81 | 22.75 |
| | 18 | 8.0090 | 0.4733 | −0.4007 | 3.5298 | 3.88 | 4.57 | 5.70 | 8.01 | 11.88 | 16.53 | 22.31 |
| | 18.5 | 7.9196 | 0.4702 | −0.3983 | 3.5766 | 3.86 | 4.53 | 5.64 | 7.92 | 11.73 | 16.25 | 21.81 |
| | 19 | 7.8366 | 0.4643 | −0.3943 | 3.6102 | 3.84 | 4.51 | 5.60 | 7.84 | 11.55 | 15.90 | 21.17 |
| | 19.5 | 7.7663 | 0.4572 | −0.3887 | 3.6266 | 3.84 | 4.50 | 5.57 | 7.77 | 11.37 | 15.55 | 20.53 |

To our knowledge, only one study employed the GAMLSS method (*Rigby & Stasinopoulos, 2004*) like we did to model SFT percentiles. The authors of this multicentre European study derived SFT percentile curves for 18,745 children ages 2–10 years but excluded overweight, obese, and underweight children from the analysis (*Nagy et al., 2014*). Thus, a direct comparison of their findings with ours is not feasible. The LMS method (*Cole & Green, 1992*) has become the most popular choice for modeling percentiles curves for anthropometric measures due to its ease of use, adoption by the World Health Organization (*De Onis et al., 2009*), and the availability of a simple software tool (LMSchartmaker, Harlow Healthcare, UK) to generate the curves. In a recent analysis of the same sample of children, we generated percentile curves for BMI, waist circumference, waist-to-height

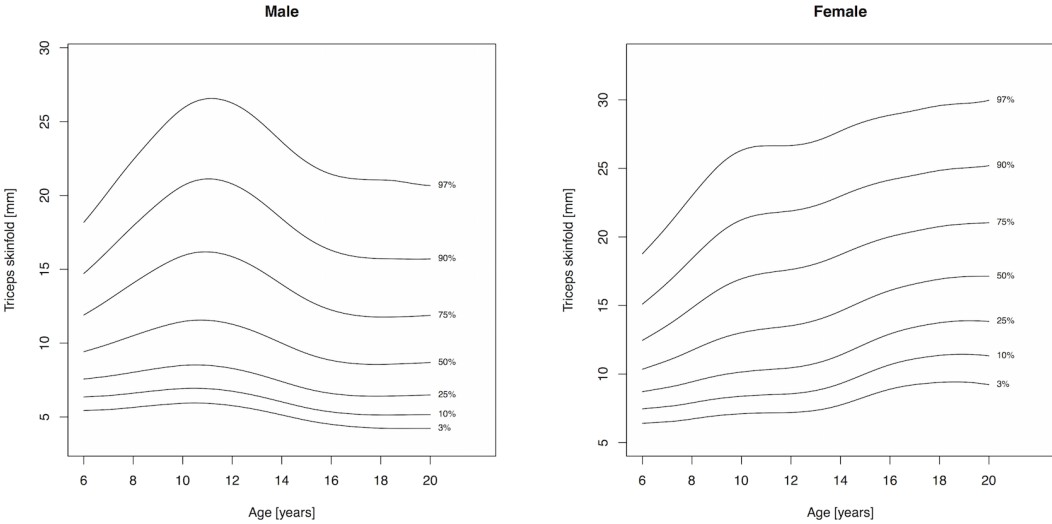

**Figure 2** Percentile curves for triceps skinfold thickness for male and female Canadian children and youth aged 6–19 years.

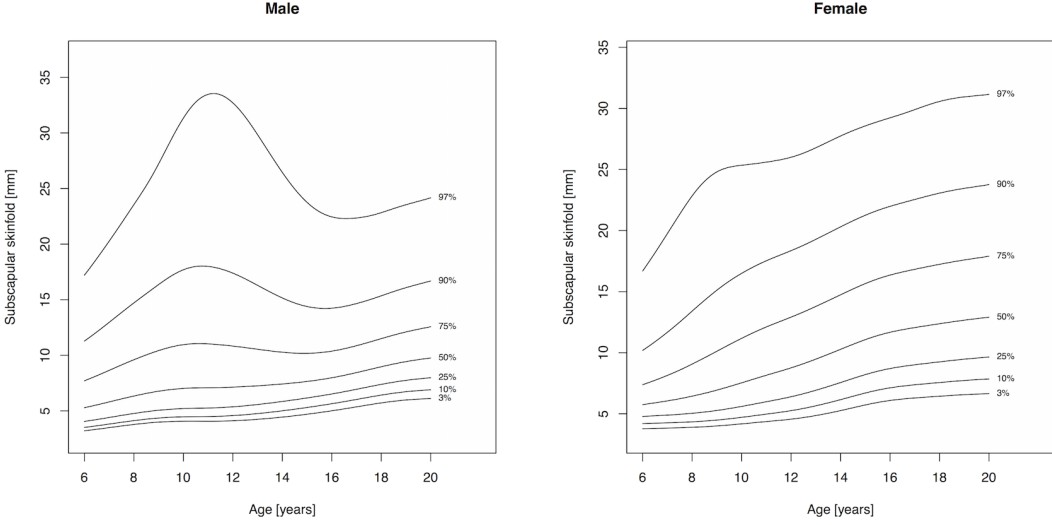

**Figure 3** Percentile curves for subscapular skinfold thickness for male and female Canadian children and youth aged 6–19 years.

ratio, and sum of five skinfolds with an adaequate model fit using the LMS method (*Kuhle et al., 2015*). However, when using the 3-parameter LMS method for the individual SFT measurements in the present study, the diagnostic worm plots revealed a large amount of kurtosis present for some variables. The LMS method attempted to account for the kurtosis with skewness, which lead to a poorer model fit at the tail end of the distribution. By contrast, the GAMLSS method includes a 4th parameter to allow the explicit modeling of kurtosis as a function of age. Diagnostics showed no model inadequacies when the curves were constructed using the GAMLSS method. Future studies should consider using the GAMLSS method if the model fit using an LMS approach is not adaequate.
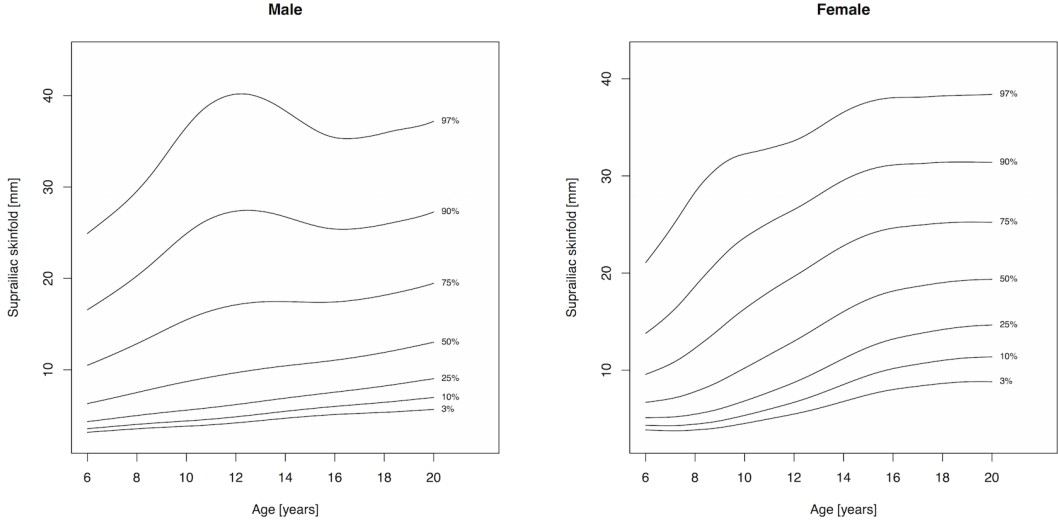

**Figure 4  Percentile curves for suprailiac skinfold thickness for male and female Canadian children and youth aged 6–19 years.**

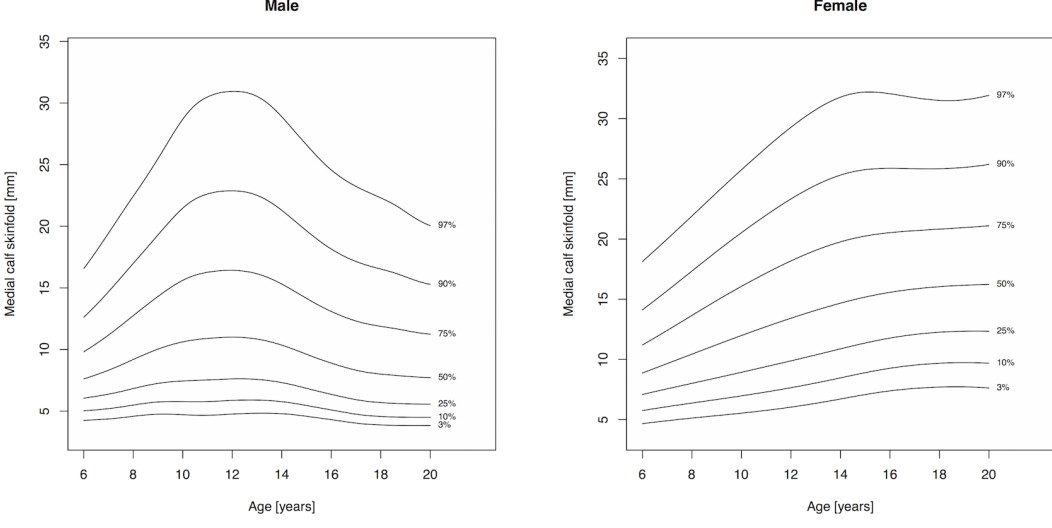

**Figure 5  Percentile curves for medial calf skinfold thickness for male and female Canadian children and youth aged 6–19 years.**

The strengths of the current study include the nationally representative sample of children and youth, and the use of sample weighting to account for non-response and design effect. The availability of a wide age range in the CHMS study population allowed us to visualize growth related trends that were not apparent in studies with narrower age ranges (*Moreno et al., 2007*; *Haas, Liepold & Schwandt, 2011*; *Brannsether et al., 2013*). We did not exclude overweight or obese children as the objective of the present study was to describe body fatness measures in a representative population of Canadian children rather than to attempt to describe what may constitute normal percentile values. Due to the physical burden of the assessments used in the survey, and the need to travel to the mobile examination clinics, there may have been a self-selection toward more mobile,

healthier, and fitter individuals. Our study is limited by the relatively small sample size, and the cross-sectional nature of the data; longitudinal data may more accurately reflect how body fatness changes with age. The omission of SFT measurements in children with a BMI greater than 30 resulted in an exclusion of 4% of children, which may have resulted in a slight downward shift of the percentiles compared to the full sample. While the flexibility of the GAMLSS method is a notable strength, its flexibility also means that the curves may differ considerably based on the parameter choices made by the researcher.

This study has presented percentile curves for SFT in a representative sample of Canadian children and youth. Since we did not examine any relationships with health outcomes or disease markers, the data should be considered as a reference for future studies and not as a growth standard.

## ACKNOWLEDGEMENTS

The analysis presented in this paper was conducted at the Atlantic Research Data Centre, which is part of the Canadian Research Data Centre Network (CRDCN). The services and activities provided by the Atlantic Research Data Centre are made possible by the financial or in-kind support of the SSHRC, the CIHR, the CFI, Statistics Canada, and Dalhousie University. The views expressed in this paper do not necessarily represent the views of the CRDCN or its partners.

### Funding

This work was supported by an IWK Health Centre (http://www.iwk.nshealth.ca) Establishment Grant awarded to Dr Stefan Kuhle (#09020) and an IWK Health Centre Research Associate Award awarded to Dr Jillian Ashley-Martin (#18396). The funders had no role in study design, data collection and analysis, decision to publish, or preparation of the manuscript.

### Grant Disclosures

The following grant information was disclosed by the authors:
IWK Health Centre Establishment: #09020.
IWK Health Centre Research Associate Award: #18396.

### Competing Interests

The authors declare there are no competing interests.

### Author Contributions

- Stefan Kuhle and Bryan Maguire conceived and designed the experiments, analyzed the data, wrote the paper, prepared figures and/or tables, reviewed drafts of the paper.
- Jillian Ashley-Martin conceived and designed the experiments, wrote the paper, reviewed drafts of the paper.
- David C. Hamilton conceived and designed the experiments, analyzed the data, reviewed drafts of the paper.

## Human Ethics

The following information was supplied relating to ethical approvals (i.e., approving body and any reference numbers):

All processes used for cycles 1 and 2 of the CHMS were reviewed and approved by the Health Canada Research Ethics Board to ensure that internationally recognized ethical standards for human research were met and maintained. Written informed consent was obtained from all participants aged 14 years and older; parents or guardians gave consent on behalf of children aged 6–13 years, while the child provided his or her assent to participate. The current project was approved by the IWK Health Centre Research Ethics Board, Halifax, NS, Canada (File # 1014413).

## Data Availability

Data are only available through the Statistics Canada Research Data Centres for researchers who meet the criteria for access to confidential data.

The application process is described at http://www.statcan.gc.ca/eng/rdc/process.

Researchers submit an application form and project proposal to the Statistics Canada Research Data Centres Program. Upon approval they have to undergo a security check. Once completed they get access to one of the Research Data Centres in Canada to analyze the data. All output produced at the centres must be vetted by a Statistics Canada analyst before it is released to the researcher for publication.

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
