# Peer review of "Percentile curves for skinfold thickness for Canadian children and youth"

_PeerJ, doi:10.7717/peerj.2247_

## Round 0.1 · original submission · Minor Revisions

· Academic Editor

Minor Revisions

To improve the paper please follow the suggestions of both reviewers.

·

Basic reporting

This is a well presented paper on an important topic, and I have no major concerns with the basic reporting.

Experimental design

The experimental design regarding the raw data collection, and the statistical modelling, are all satisfactory. I have greater reservations over the conversion of the raw skinfold data to percentage fat, as several publications have demonstrated that the Slaughter equations have substantial bias (eg Wells et al., AJCN 1999; 69(5):904-12). Furthermore, there are now reference data for body fat measured directly by criterion methods (eg Wells et al., AJCN 2012; 96(6):1316-26).

Validity of the findings

All the findings relating to the skinfolds themselves appear valid. Again, however, I have reservations about presenting predictions of body fat using equations already considered to be biased.

Additional comments

In general this is an excellent paper, providing new and robust skinfold thickness reference data for the Canadian population over the majority of the paediatric age range. I am well aware that skinfolds are often converted into percentage fat data, using published equations. This is a problematic approach, as no skinfold equation has yet been published that transfers successfully across populations. Inevitably, the product of these equations is bias, and this bias is likely to differ between populations. Therefore, although this approach might appear to enable comparison of adiposity across populations, this approach is to my mind flawed.

Furthermore, the authors make no reference to published data on fat mass, or even fat mass index, using criterion methods such as isotope dilution, the four component model, DXA etc (eg Wells et al,. AJCN 2012). Given these much more robust data on tissue masses, predictions based on skinfold measurements do not contribute comparable objective data. My preference would be that these body fat predictions are dropped from the manuscript, because it is very valuable for clinicians to rank adiposity using the raw skinfold data anyway. For a discussion of the limitations of using prediction equations, see Wells and Fewtrell, Arch Dis Child 2006; 91(7):612-7.

It would also be helpful for the authors to add a qualification that skinfold thicknesses do not assess internal visceral adiposity, which has the strongest association with ill-health. In general, there are fairly good correlations between skinfold thicknesses and internal adiposity, nevertheless this limitation should be acknowledged and discussed.

Reviewer 2 ·

Basic reporting

This study is about a topic with important practical implications. Although skinfold thickness is used widely as a proxy of body fat, very few data exist on large populations. The findings of this study are expected to help health-related practitioners to evaluate body fat in children. Based on the importance of this study I recommend it for publication with minor corrections.

1) The title does not reflect accurately the content. Either delete the distinction in peripheral and truncal from the title or add more information in the Introduction and Discussion about this distinction.

2) l.31: girls in girls: curves in girls

3) Intoduction: Why the particular five skinfolds have been chosen? (e.g. popularity, Eurofit…)

4) Basic details are needed in the Methods: how age, weight and height were assessed?

Experimental design

No Comments

Validity of the findings

No Comments

---

## Round 0.2 · accepted · Accept

· Academic Editor

Accept

The manuscript has been improved following the reviewers suggestions.